# Evaluation of Maternal and Neonatal Outcomes of Assisted Reproduction Technology: A Retrospective Cohort Study

**DOI:** 10.3390/medicina56010032

**Published:** 2020-01-15

**Authors:** Hiroaki Tanaka, Kayo Tanaka, Kazuhiro Osato, Hideto Kusaka, Yuka Maegawa, Haruki Taniguchi, Tomoaki Ikeda

**Affiliations:** Department of Obstetrics and Gynecology, Mie University School of Medicine, 2-174 Edobashi, Tsu, Mie 514-0008, Japan; tagami.t.ky@gmail.com (K.T.); karemakosato@gmail.com (K.O.); irc_sankalist@ise.jrc.or.jp (H.K.); yuka.ma.0513@gmail.com (Y.M.); vzd01336@nifty.com (H.T.); t-ikeda@clin.medic.mie-u.ac.jp (T.I.)

**Keywords:** maternal fetal medicine, assisted reproductive technology, maternal outcomes, neonatal outcomes

## Abstract

*Background:* To evaluate maternal and neonatal outcomes of assisted reproductive technology (ART). *Materials and Methods:* Pregnant women registered from 2015 through 2017 (*n* = 6994) at five perinatal centers that managed high-risk pregnancies in Mie, Japan, retrospectively. Rates of preterm birth (<37 gestational weeks), early onset preeclampsia (<34 gestational weeks), late onset preeclampsia (≥34 gestational weeks), low-lying placenta, placenta previa, placenta accreta, placental abruption, atonic bleeding, uterine rupture, and amniotic fluid embolism after ART were evaluated. ART was defined as in vitro fertilization and micro-fertilization. Fisher’s exact test, Mann–Whitney’s U test, and logistic regression analysis were used to analyze the data. *Results:* Rates of obstetrical complications including low-lying placenta, placenta previa, placenta accreta, and atonic bleeding were increased with ART compared to those with the control. Particularly, ART was associated with a significantly increased rate of placenta accreta (adjusted odds ratio: 7.35, 95% confidence interval (CI): 3.20–16.6) and significantly decreased rate of placental abruption (adjusted odds ratio: 0.24, 95% CI: 0.07–0.61). *Conclusions:* This study showed that ART may reduce placental abruption and increase placenta previa. There is a possibility that the placenta attaches deeper in the myometrium because of ART.

## 1. Introduction

Assisted reproductive technology (ART) has advanced significantly in recent years, and the incidence of pregnancy by ART has increased. Currently in Japan, one in 18 women give birth because of ART [1]. The number of babies born by ART is also predicted to increase in the future. Therefore, perinatal outcomes for pregnancy by ART need to be understood. Currently, pregnancies by ART are known to be associated with a high rate of premature birth, low-birth-weight infants, fetal malposition, abnormal placenta and umbilical cord, high rate of cesarean section, perinatal mortality, and high amount of blood loss during delivery [2,3,4,5,6,7,8]. However, perinatal outcomes for pregnancy by ART are not yet understood in their entirety. This study used data accumulated in perinatal centers that manage high-risk pregnancies to evaluate maternal and neonatal outcomes of pregnancies by ART.

## 2. Materials and Methods

Between 2015 and 2017, we screened every pregnancy and delivery managed in every perinatal center in the Mie Prefecture of Japan (five facilities) that managed high-risk pregnancies.

### 2.1. Data Collection

The use of ART was surveyed as a maternal background characteristic from medical records, and pregnancies were grouped as follows: the control group included natural pregnancy, artificial insemination, and ovulation induction, whereas the ART group included pregnancy by in vitro fertilization and micro-fertilization. Age of the mother, number of parity, number of weeks before delivery, number of vaginal deliveries, and number of cesarean deliveries, and additional information including the ratio of frozen embryo transfer and frozen embryo transfer as well as the ratio of natural and hormonal cycle were collected. Surveyed obstetrical complications included preterm birth (<37 gestational weeks), early onset preeclampsia (<34 gestational weeks), late onset preeclampsia (≥34 gestational weeks), low-lying placenta, placenta previa, placenta accreta, placental abruption, atonic bleeding, uterine rupture, and amniotic fluid embolism. Surveyed obstetrical interventions included emergency cesarean section, maternal transfusion, hysterectomy, and manual removal of the placenta. Surveyed neonatal outcomes included birth weight, large for gestational date, small for gestational date, Apgar score <7 (at 1 minute), and Apgar score <7 (at 5 min). This study was approved by the Ethics Committee at the Mie University Hospital of Japan (receipt number t2018-4, date of approval 28 December 2018).

### 2.2. Statistical Analysis

Fisher’s exact test and Mann–Whitney’s U test were used to calculate the P-values for the comparison of maternal characteristics. Logistic regression analysis was used to calculate the adjusted odds ratios (aORs) and 95% confidence intervals (CIs) in the analysis of adverse pregnancy outcomes. Odds ratios were adjusted by age and parity. All statistical analyses were performed using JMP PRO 11 software (SAS Institute Inc., Cary, NC, USA).

## 3. Results

We screened 6994 cases, and 42 cases with insufficient information were excluded. Ultimately, 6141 cases were included in the control group and 811 cases were included in the ART group (Figure 1). 

Table 1 shows the maternal background characteristics. Maternal age, number of primipara, and rate of cesarean delivery were significantly higher in the ART group than in the control group. In the ART group, there were 730 cases of frozen embryo transfer (90%) and 795 cases of hormone cycle therapy (98%).

Table 2 shows the obstetrical complications. Rates of low-lying placenta, placenta previa, placenta accreta, and atonic bleeding were increased in the ART group compared with those in the control group. In particular, the rate of placenta accreta increased significantly (control group: 18 cases (0.3%) vs. ART group: 18 cases (2.6%), aOR: 7.35, 95% CI: 3.20–16.6, *P* < 0.001). However, placental abruption decreased significantly in the ART group compared with that in the control group (103 cases (1.7%) vs. 4 cases (0.5%), aOR: 0.24, 95% CI: 0.07–0.61, *P* < 0.001). The amounts of blood loss during delivery were 662.1 ± 6.8 mL in the control group and 998.2 ± 18.9 mL in the ART group, and the amount increased significantly in the latter group.

Regarding obstetrical interventions, rates of emergency cesarean section, maternal transfusion, and manual removal increased significantly in the ART group compared with the control group (Table 3). Birth weight was not significantly different in both groups (2763 ± 8.0 vs. 2782 ± 22.2, *P* = 0.41). Concerning neonatal outcomes, rates of large for gestational age, Apgar score < 7 (at 1 min), and Apgar score < 7 (at 5 min) were significantly higher in the ART group than in the control group (Table 4).

## 4. Discussion

This study suggests that placental abruption decreases and placenta accreta increases with ART. Results of the other obstetrical complications and neonatal outcomes were similar to those in previous reports except for decreases in placenta abruption [2,3,4,5,6,7,8].

ART may cause excessive attachment of the placenta to the myometrium and formation of a placenta that is difficult to peel off. Various causes have been speculated including previous intrauterine operation and thinning of the intrauterine membrane by hormone replacement such as estrogen and progesterone. With regard to excessive attachment of the placenta [9], the pathology remains unknown and is therefore only a speculation. Investigating such causes was difficult in the present study. Therefore, elucidation of the increase in placenta accreta and placental abruption due to ART on a molecular basis is needed in the future. It is also important to note that most women in the ART group underwent frozen embryo transfer and hormone cycle therapy. The rate of placenta accreta is known to increase because of frozen embryo transfer [9], so previous intrauterine operation in women with ART may not be the only cause.

Regarding obstetrical interventions, manual removal of the placenta increased, but this was a reasonable outcome considering the increase in placenta accreta. A significant increase of maternal transfusion in the ART group is speculated to be due to an increase in blood loss during delivery caused by factors such as an increase in cesarean sections. However, the reasons behind the increase in cesarean sections in the ART group could not be revealed in this study.

The age and primipara rate in the ART-group were higher than in the control group. Elderly primigravida is the risk factor of caesarean deliveries. Therefore, the rate of caesarean delivery is higher in the ART-group.

Concerning neonatal outcomes, the incidence of a high-fat diet (HFD) has increased, as reported in previous studies. The reason behind the increase in HFD has yet to be identified. Therefore, elucidation of the increase in HFD on a molecular basis is needed in the future. Long-term follow-up of children with HFD born by ART is also required.

This study is limited by the fact that the accumulated perinatal data were from facilities that managed mainly high-risk pregnancies; therefore, many of the pregnancy background characteristics are representative of high-risk pregnancies. Few women with low-risk pregnancies were included in the control group. This may be the reason why there was no difference in the premature delivery rate, unlike in past reports. Another limitation of this study was that it was conducted retrospectively.

## 5. Conclusions

This study showed that ART may reduce placental abruption. The decrease in the number of cases of placental abruption differs from past reports; thus, reconsideration of ART is necessary. There is also a need to further understand the relationship between ART and perinatal complications (i.e., the actual conditions and pathology), which are expected to increase in the future.

## Figures and Tables

**Figure 1 medicina-56-00032-f001:**
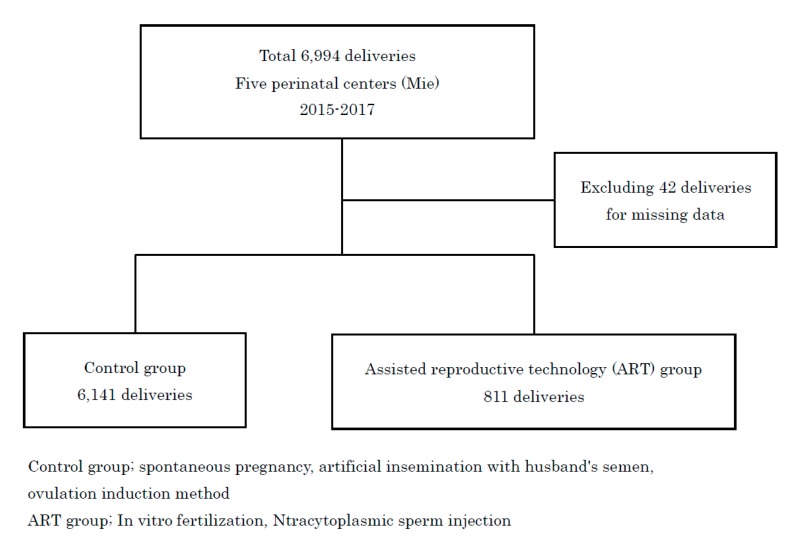
Study design.

**Table 1 medicina-56-00032-t001:** Maternal background.

Maternal Background Characteristics	Control Group (*n* = 6141)	ART Group (*n* = 811)	*P*-Value
Maternal age (year)	31.0 ± 0.2	37.2 ± 0.1	<0.001
Primipara	2956 (47.9%)	556 (68.6%)	<0.001
Gestational weeks at birth (weeks)	37.5 ± 0.1	37.4 ± 0.1	0.82
Delivery methods			
Vaginal delivery	3564 (58.0%)	318 (39.2%)	<0.001
Caesarian delivery	2577 (42.0%)	493 (60.1%)	
Fresh embryo transfer		81 (10.0%)	
Frozen embryo transfer		730 (90.0%)	
Natural cycle		16 (2.0%)	
Hormone cycle		795 (98.0%)	

**Table 2 medicina-56-00032-t002:** Obstetrics complication.

Obstetrics Complication	Control Group (n = 6141)	ART Group (n = 811)	aOR	95% CI	*P*-Value
Preterm birth	1274 (21.5%)	178 (22.5%)	1.01	0.81–1.24	0.92
Early onset preeclampsia	79 (1.3%)	13 (1.6%)	1.19	0.64–2.34	0.59
Late onset preeclampsia	252 (4.1%)	57 (7.0%)	1.17	0.83–1.63	0.34
Low lying placenta	79 (1.3%)	21 (2.6%)	1.83	1.04–3.12	0.03
Placenta previa	96 (1.6%)	34 (4.2%)	1.96	1.24–3.06	0.004
Placenta accreta	18 (0.3%)	18 (2.6%)	7.35	3.20–16.6	<0.001
Placental abruption	103 (1.7%)	4 (0.5%)	0.24	0.07–0.61	0.001
Atonic bleeding	103 (1.7%)	31 (3.8%)	1.95	1.22–3.07	0.006
Uterine rupture	4 (0.006%)	1 (0.1%)	3.09	0.13–30.16	0.40
Amniotic fluid embolism	3 (0.005%)	2 (0.2%)	5.93	0.59–55.43	0.12

ART—assisted reproductive technology; aOR—adjusted odds ratio; CI—confidence interval.

**Table 3 medicina-56-00032-t003:** Obstetrics intervention.

Obstetrical Interventions	Control Group (*n* = 6141)	ART Group (*n* = 811)	aOR	95% CI	*P*-Value
Emergency caesarian section	1003 (16.2%)	195 (24.4%)	1.23	1.01–1.49	0.03
Maternal transfusion	122 (2.0%)	42 (5.2%)	2.04	1.34–3.04	0.009
Hysterectomy	7 (0.001%)	1 (0.001%)	0.57	0.08–11.95	0.65
Manual removal	79 (1.3%)	29 (3.6%)	2.68	1.62–4.37	<0.001

ART—assisted reproductive technology; aOR—adjusted odds ratio; CI—confidence interval.

**Table 4 medicina-56-00032-t004:** Neonatal outcome.

Neonatal Outcome	Control Group (*n* = 6141)	ART Group (*n* = 811)	aOR	95% CI	*P*-Value
Large for gestational date	91 (1.5%)	24 (3.4%)	1.90	1.10–3.19	0.004
Small for gestational date	151 (2.4%)	18 (2.2%)	0.95	0.54–1.61	0.80
Apgar score < 7 (1 min)	710 (11.7%)	130 (16.0%)	1.50	1.19–1.89	<0.001
Apgar score < 7 (5 min)	206 (3.3%)	46 (5.6%)	1.30	1.22–2.61	0.001

ART—assisted reproductive technology; aOR—adjusted odds ratio; CI—confidence interval.

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
