# Peer review of "Evaluation of Maternal and Neonatal Outcomes of Assisted Reproduction Technology: A Retrospective Cohort Study"

_medicina, 2020, doi:10.3390/medicina56010032_

Round 1

Reviewer 1 Report

Evaluation of maternal and neonatal outcomes of assisted reproduction technology: a retrospective cohort study

In this paper Dr. Tanaka and colleagues report on a retrospective centre-based study to evaluate the maternal and neonatal outcomes of 6,994 pregnant women registered from 2015 through 2017 in five perinatal centres.

The paper itself is well-written, and the findings are described meticulously and honestly.

However, there are some general weaknesses of the report:

There is no comment on multiples. This is essential in ART.

There is no comment on the fact, that perhaps pregnant women had inhibitions to declare, that the pregnancy came from an Art-treatment.

There is no link between birth-registry and ART-registry like in other counties.

There should be a comment on the higher rate of Caesarean deliveries in the ART-group.

The definition of the control group: natural pregnancy, artificial insemination, and ovulation induction.

One finding is: This study suggests that placental abruption decreases and placenta accreta increases with ART.

In the discussion it is mentioned:  Various causes have been speculated including previous intrauterine operation and thinning of the intrauterine membrane by hormone replacement such as  estrogen and progesterone.

This also can happen in pregnancies after artificial insemination and ovulation induction.

So this seems to be a weak argument.

There should be a comment on this.

The paper should be reviewed again in a revised version.

Author Response

Reviewer 1

December 24th, 2019

Medicina

Dear Reviewer

We wish to re-submit the manuscript titled “T Evaluation of maternal and neonatal outcomes of assisted reproduction technology: a retrospective cohort study.” The manuscript ID is medicina-608577. We greatly appreciate the reviewers’ constructive comments.

We hope that the revised manuscript addresses all reviewer concerns and is suitable for publication in the journal. Thank you for your consideration. I look forward to hearing from you.

There should be a comment on the higher rate of Caesarean deliveries in the ART-group.

The age and primipara rate in ART-group is higher than Control group. Generally, elderly primigravida is the risk factor of caesarean deliveries. We have added the sentence in the discussion section as suggested by you.

In the discussion it is mentioned: Various causes have been speculated including previous intrauterine operation and thinning of the intrauterine membrane by hormone replacement such as estrogen and progesterone.

This also can happen in pregnancies after artificial insemination and ovulation induction. So this seems to be a weak argument.

Thank you for your comment. We agree your suggestion. However, the reason increasing the rete of placenta accrete is not understood. We have to speculate including previous intrauterine operation and thinning of the intrauterine membrane by hormone replacement such as estrogen and progesterone. We try to clarify the reason increasing the caesarian delivery in nest basic and clinical study

Sincerely,

Hiroaki Tanaka, MD, PhD

Department of Obstetrics and Gynecology, Faculty of Medicine, Mie University

2-174 Edobashi, Tsu, Mie, Japan.

Telephone: +81-059-232-1111; Fax: +81-059-231-5202

Reviewer 2 Report

Well written and presented article. The authors have succeeded to produce a nice manuscript , its length is not extensive and the results are presented clearly and the methodology is clear. However the conclusions are much known to the readers. In conclusion this paper offers known results in a nice scientific and short way, 

Author Response

Reviewer 2

December 24th, 2019

Medicina

Dear Reviewer

We wish to re-submit the manuscript titled “T Evaluation of maternal and neonatal outcomes of assisted reproduction technology: a retrospective cohort study.” The manuscript ID is medicina-608577. We greatly appreciate the reviewers’ constructive comments.

We hope that the revised manuscript addresses all reviewer concerns and is suitable for publication in the journal. Thank you for your consideration. I look forward to hearing from you.

Sincerely,

Hiroaki Tanaka, MD, PhD

Department of Obstetrics and Gynecology, Faculty of Medicine, Mie University

2-174 Edobashi, Tsu, Mie, Japan.

Telephone: +81-059-232-1111; Fax: +81-059-231-5202
